# A Study on the Preliminary Validity Analysis of Korean Firefighter Job-Related Physical Fitness Test

**DOI:** 10.3390/ijerph19052587

**Published:** 2022-02-23

**Authors:** Eun-Hyung Cho, Jung-Hoon Nam, Seung-A. Shin, Jong-Back Lee

**Affiliations:** 1Department of Sports Science, Korea Institute of Sport Science, Seoul 01794, Korea; ehcho@kspo.or.kr; 2Department of Sports Healthcare, Catholic Kwandong University, Gangneung 25601, Korea; n7j7h782@knsu.ac.kr; 3Department of Sports and Leisure Industry, Sehan University, Dangjin 31746, Korea; ssa0618@sehan.ac.kr; 4Center for Sports Science in Gangwon, Chuncheon 24239, Korea

**Keywords:** ${\dot{V}O}_{2\max}$, firefighter job-related physical fitness test, candidate physical ability test, anaerobic power

## Abstract

The purpose of this study is to revise and modify the firefighter job-related performance tests from overseas to implement into the circumstances in Korea, examine its validity by analyzing the level of association between the test employed in the ongoing firefighter selection process, and propose a Korean firefighter job-related physical fitness test. Therefore, a modified version of Candidate Physical Ability Test (CPAT) from the United States firefighter selection process was conducted on 28 male firefighter officer candidates. Recorded results from VO_2_max, heart rate, blood lactate, anaerobic power test, and the ongoing Korean firefighter physical fitness test were analyzed to assess the validity of the modified test. IBM SPSS Statistics Ver. 27.0 was employed for the data correlation analysis in different fitness factors and the total circuit physical test time. The results revealed the proposed modified firefighter job-related physical fitness test showed significant correlation with VO_2_max (r = −0.450, *p* < 0.05), METS (r = −0.735, *p* < 0.01) recovery lactate over 15 min (r = −0.460, *p* < 0.05), peak power (r = −0.484, *p* < 0.05), average power (r = −0.647, *p* < 0.01), and in the ongoing firefighter physical fitness test, grip strength (r = −0.709, *p* < 0.01), lower back strength (r = −0.681, *p* < 0.01), standing long jump (r = −0.618, *p* < 0.01), sit-ups (r = −0.397, *p* < 0.05), and shuttle run (r = −0.523, *p* < 0.01). Fitness factors including VO_2_max, recovery lactate, anaerobic power, muscular strength, and so forth known to play a crucial role in firefighting operations were also shown to be important in the modified firefighter job-related physical fitness test. However, we suggest that studies with a larger sample size are needed in order to generalize our findings.

## 1. Introduction

Firefighters bear a higher injury rate compared to other specialized occupations as their duties in conducting fire suppression, rescue, and emergency first-aid operations occur in hazardous environments. A total of 1789 injuries (367 firefighting, 161 rescue, 400 first-aid, 25 educational training, 11 deaths, 825, etc.), were reported in 3 years from the Deceased and Injured Korean Firefighter statistics between 2016 and 2018 [1]. Due to the hazardous work environment, the firefighters carry about 22 kg of individual protective gear and self-contained breathing apparatus (SCBA), and participate in different firefighting and rescue operations. A high level of muscular strength and endurance is required in order to properly perform firefighting and rescue operations with heavy equipment, and based on a previous study on the physical exertion in firefighting operations, the maximal heart rate reaches up to 84–100%, and the peak oxygen uptake to 63–97% [2]. Furthermore, it is accompanied by anaerobic metabolism and greater lactic acid concentration of 6–13 mmol/L, which is known to accumulate temporarily during high intensity physical activities [3].

Therefore, excellent physical fitness is required for firefighters considering the high extrinsic job risk and physical demands. The Ongoing National Occupational Firefighter Examination assesses six different basic physical fitness categories that are deducted from the study by Ko [4], based on the Firefighter Candidate Physical Ability test by the National Fire Protection Association (NFPA), which was established in a study by Williford et al. [5]. However, this ongoing examination on current firefighters and firefighter candidates is recognized as insufficient for a proper assessment of the requirements for firefighting operations evidenced by low physical fitness standards and simple fitness tests. According to a study that compared physical fitness levels in firefighters between Korea and other countries, Korean firefighters’ overall fitness levels were reported to be inferior [6].

In most developed countries such as the United States, Australia, or the United Kingdom, job-related performance testing is conducted either individually or accompanied by a physical fitness test in the firefighter selection process. Although the physical fitness tests that are specific to the occupational characteristics of firefighters are regarded to bear long-term positive effects as they contribute to job-related performance improvement, injury prevention, and development in firefighting operation, relevant studies on the physical fitness test and examination on Korean firefighters remain insufficient.

The ongoing firefighter selection process is restricted in terms of test battery choice owing to the limitation of equipment, time, and human resources available while having to mass test a large number of candidates concurrently. Moreover, the minimal criterion and test battery can vary depending on which principle is chosen over empirical, task analysis, health criterion, age and gender, trainability, and so forth. The purpose of this study is to revise the firefighter job-related performance tests from overseas, evaluate its validity by comparison with the ongoing physical fitness test and analyze the level of association with the job performance to propose a Korean-model firefighter physical fitness test.

## 2. Materials and Methods 

### 2.1. Data Collection

A modified job-performance physical fitness test was conducted on a total of 28 male firefighter officer candidates registered in the National Fire Service Academy in 2020. The test was conducted once the study purpose and procedure were thoroughly explained to the candidates, and provided consent to the study, and then cleared to participate with the Physical Activity Readiness Questionnaire. The anthropometric characteristics of subjects are represented in Table 1.

### 2.2. Korean-Model Firefighter Job-Related Physical Fitness Test

The Korean-model firefighter job-related physical fitness test is a modified version of the United States’ firefighter selection test or, Candidate Physical Ability Test (CPAT), and was adapted to fit the conditions in Korea.

CPAT was originally developed to manage the physical fitness of firefighters in 1997 and can screen whether a new firefighter exhibits the requisite physical ability to perform basic duties at a fire scene. The test battery is comprised of a total 8 different assessments (Stair climb, Hose pull, Equipment carry, Ladder Raise, Forcible entry, Search, Victim drag, Ceiling breach and pull) that has to be performed in a continuous manner within the time limit of 10 min 20 s while wearing a 50-pound vest. CPAT was established to hire competent candidates who could carry out the essential duties and operations at a fire scene. As Gledhill’s paper [7] identifies, CPAT includes the firefighters’ typical movements patterns such as climbing, pushing, pulling, lifting, and chopping, and also shares similarity with the Firefighting Performance Contest governed by the U.S NFPA, and Korean Firefighting Skill Competition.

Various limitations are expected in adopting CPAT as a whole as few intrinsic differences exist in the firefighter selection process between the U.S and Korea. Thus, we modified the test considering its distinguishability, suitability, reliability, objectivity, measurements’ accessibility, and the safety of participants.

The U.S CPAT is operated by each fire department at different authorized testing sites with standardized instruments. However, the need for mass testing on a large number of candidates in a short amount of time imposes limitations when practiced in Korea without any modification on the test battery or procedure. Furthermore, the modified test should be able to distinguish the different levels of physical fitness between applicants as the physical fitness test comprises 15% of the total score after a written test in the selection process. Based on these limitations, we adjusted the grading system (CPAT—Pass or non-Pass, Modified—Total time taken), and a part of the test battery (Table 2). If we observe the difference, the stair climb in CPAT assesses both lower body muscular strength and aerobic fitness of the firefighter applicant. On the other hand, due to the limitation of mass testing in a short amount of time, and the applicant’s accessibility to practice beforehand of the examination, the stair climb was exchanged with the burpee test. 

The burpee test measures muscular strength and cardiovascular endurance and was set at 20 repetitions in consideration of its influence on other forthcoming assessments [8]. In the equipment carry exercise, the applicant has to move 20 m carrying two 20 kg items, a total of 40 kg. This is referred to the weight of the equipment used in Korea based on ‘2020 Firefighting Equipment Standard Specification’ whereas the firefighters generally handle gear such as fire extinguishers, ladders, or rescue equipment. 

The pulling motion represents a typical movement that sees practical usage in firefighting and is also mainly used in the CPAT’s forcible entry, ceiling breach and pull [7]. Our study adopted alternative tests such as the medicine ball throw, inverted row, sledge push, and others that activate the same muscle groups [9]. The height of the medicine ball throw was determined considering the weight of thrust force and its potential energy, while the weight of sledge in the sledge push considered a water shooting pressure of 2.5 MPA (25 kg) exerted from the hose [6]. The modified firefighter physical fitness test proposed by our study is illustrated in Table 3.

**Table 2 ijerph-19-02587-t002:** Comparison of CPAT and modified physical fitness test [10].

CPAT	Detail	Modification	Detail
Stair Climb	Warm up: 20 s 50 step/min Velocity test: 3 min 60 step/min Velocity	Burpee Test	20 rep
Hose Pull	100 ft (30.48 m) drag, 50 ft (15.24 m) pulling	Hose Pull	9 kg hose, 20 m drag and pulling
Equipment Carry	Circular Saw 32 ± 3 lbs (14.5 ± 1.3 kg) Chain Saw 28 ± 3 lbs (12.7± 1.3 kg), 75 ft (22.86 m)	Equipment Carry	20 kg kettle bell ×2 total 40 kg, 20 m move
Ladder Raise	24 ft (7.32 m) repeat raising and lowering 2 drag ladders	Medicine Ball Throw	8 kg Toss the medicine ball to a height of 2.8 m or more
Forcible Entry	4.54 kg Use a hammer to hit the measuring instrument until a signal is received.	Inverted Row	15 rep Pull
Ceiling Breach and Pull	60 pounds (27.22 kg) 3 pushes, 80 pounds (36.29 kg) 5 pulls	Sledge Push	When pushed, the sled weighs 25 kg and moves 20 m.
Search	Pass through a structure consisting of 91.44 cm high, 4 feet (121.92 cm) wide, and 64 feet (19.51 m) long, 20 ft-24–20 ft long.	Tunnel get through	Pass through 20 m of a structure with a square width of 90 cm up and down.
Victim Drag	The 74.84 kg mannequin returns a distance of 10.67 m.	Victim Drag	The 74.84 kg Victim returns a distance of 20 m.
		400 m Shuttle Run	It makes 20 round trips for a distance of 20 m.

The burpee test measures muscular strength and cardiovascular endurance and was set at 20 repetitions in consideration of its influence on other forthcoming assessments [8]. In the equipment carry exercise, the applicant has to move 20 m carrying two 20 kg items, a total of 40 kg. This is refered to the weight of the equipment used in Korea based on ‘2020 Firefighting Equipment Standard Specification’ whereas the firefighters generally handle gear such as fire extinguishers, ladders, or rescue equipment. 

The pulling motion represents a typical movement that sees practical usage in firefighting and is also mainly used in the CPAT’s forcible entry, ceiling breach and pull [7]. Our study adopted alternative tests such as the medicine ball throw, inverted row, sledge push, and others that activate the same muscle groups [9]. The height of the medicine ball throw was determined considering the weight of thrust force and its potential energy, while the weight of sledge in the sledge push considered a water shooting pressure of 2.5 MPA (25 kg) exerted from the hose [6]. The modified firefighter physical fitness test proposed by our study is illustrated in Figure 1.

#### 2.2.1. The Battery and Procedure

To assess the fitness factors of the modified firefighter physical fitness test, a portable blood gas analyzer (Cosmed K5, Rome, Italy) was used to examine the relationship between the maximal oxygen uptake (VO_2_max), and the Wingate test was conducted to examine anaerobic power. The list of measurements is shown in Table 3.

#### 2.2.2. Korea Firefighter Physical Fitness Test

(1)The muscular strength is determined by measuring lower back and handgrip strength with a digital dynamometer (Tkk-1270, Takei, Tokyo, Japan) in a kilogram unit. The subject exerts maximal effort, grabbing the dynamometer with the index finger flexing the interphalangeal joint at a 90-degree angle. The best record is selected out of 2 trials on each side of the hand. The lower back strength is measured with the subject holding the handle of the dynamometer leaning forward at a 30-degree angle with knees and arms extended. The best record is chosen out of 2 trials.(2)The muscular endurance is measured by sit-ups performed in 1 min. The subject holds hands behind their head knitting fingers together. A repetition is counted when both elbows touch the knee after shoulders fully contact the ground.(3)The flexibility of a subject is measured by sit and reach using WL-35 (Yagami, Nagoya, Japan). The best record is chosen out of 2 trials after warm-up stretching on related muscle groups including hamstrings and glutes.(4)The muscular power is measured by standing long jump. The subject jumps and lands using both feet, swinging arms, and bending knees in a rhythm to launch a forward drive. The longest distance of 2 trials was chosen in units of 0.1 cm.(5)The shuttle run test is conducted with our own soundtrack to signal the participants after an adequate explanation of the test procedure. The subject is given a warning if they cannot reach the 20 m distance between the signals and is eliminated after the second warning. The time immediately before elimination was recorded.

The height, weight, and body fat percentage (%fat) were obtained using the extensometer, and bioelectrical impedance analyzer (InBody-720, InBody, Seoul, Korea), and the body composition was calculated. Due to intra-daily variability in height, weight, and %fat measurements were taken between 9–11 a.m. The height was recorded in centimeters rounded off the numbers to one decimal place.

#### 2.2.3. Anaerobic Power Test

The anaerobic power of the subject was measured with the Wingate test using a cycle ergometer (Monark 828E, Vansbro, Sweden). After 2 min of warm-up with light pedaling, when the set workload is applied, the subject is signaled to accelerate maximally and maintain the speed as much as possible for 30 s. After the finishing the test, the calculated anaerobic peak power and anaerobic mean power were recorded.

#### 2.2.4. Maximal Oxygen Consumption Measurement

The best method to confirm whole body endurance, or aerobic fitness is to measure the VO_2_max of a subject. After calibration, the subject is equipped with a portable blood gas analyzer (COSMED K5, Rome, Italy) and a heart rate monitor. The discomfort during movement was minimized by applying tape and wearing a jacket to hold the analyzer close to the body.

#### 2.2.5. Lactate Measurement

Lactate is a useful biomarker created in muscles during high intensity exercise which accumulates in the blood. The blood lactate concentration is utilized in exercise planning and assessment on athletes. Our study employed the lactate analyzer (EKF Biosen C-line, Magdeburg, Germany), and collected blood lactate samples at rest, 5 min, 15 min, and immediately after the modified firefighter physical fitness test with the fingertip method. To avoid any contamination of samples or hepatitis contagion, the testers donned latex gloves with proper disinfection using an alcohol swab.

### 2.3. Statistical Analysis

All measured variables were analyzed with IBM SPSS Statistics 27 (IBM Corp: Armonk, NY, USA, 2020) to calculate the mean value and standard deviation. Pearson’s correlation coefficient was utilized to examine the correlation between variables, with the statistically significant level set at α = 0.05.

## 3. Results

### 3.1. Modified Job Related Performance Test Measurement Result

The results measured including VO_2_max by the portable blood gas analyzer, metabolic equivalent, heart rate, peak and average anaerobic power by the Wingate test, and blood lactate test are shown in Table 4.

### 3.2. Relationship of VO_2_max and Wingate Test and Lactate Recovery Ratio on Correlation Analysis

The results of the correlation analysis between the total circuit physical test time, VO_2_max, Wingate test, blood lactate levels, and others are shown in Table 5. A high correlation was detected between the total circuit physical test time, and VO_2_max, METS, and anaerobic average power (*p* < 0.01). Moreover, the lactate recovery levels and anaerobic peak power were correlated at a statistically significant level (*p* < 0.05). However, there was no significant relationship between the anaerobic power drop ratio and all-out lactate level (*p* < 0.05).

### 3.3. Relationship of Improvement Physical Fitness Test and Korea Firefighter Physical Fitness Test on Correlation Analysis

The result of correlation analysis between the modified firefighter physical fitness test and the current Korea physical fitness test are shown in Table 6. The circuit physical test time in the modified test, and grip strength, lower back strength, standing long jump, and shuttle run from the current test appeared to bear a high level of correlation (*p* < 0.01). The sit-ups from the ongoing test also correlated with the circuit test time at a statistically significant level (*p* < 0.05). However, there was no correlation with the sit and reach (*p* > 0.05).

## 4. Discussion

Quantification of the physiological capacity needed for firefighters at real-world fire scenes remain difficult and complex [11]. In few studies that reported the physiological responses of firefighters including heart rate, percentage of maximum heart rate, ratings of perceived exertions, and oxygen uptake in simulated fire scenes such as ceiling overhaul, stair climb with a high-rise pack, crawl, search and rescue, elevated level of oxygen uptake, heart rate, blood lactate concentration, and salivary cortisol and a-amylase were detected due to high psychological pressure and thermal stress, smoke, SCBA, and high intensity physical activities such as carrying 25 kg of protective gear and heavy objects or participating in a rescue operation [12]. Among several fitness factors, cardiovascular endurance and muscular strength are reported as the most important and contribute to more efficient task execution in fire-fighting. The study strongly recommended high levels of cardiovascular endurance and muscular strength in order to ensure the safety of both firefighters and victims, and were revealed to enhance the task performance and safety while decreasing the risk of injury. Likewise, the study by Michaelides et al. [13] reported a positive relationship between a high level of fitness and firefighting task performances. Among several fitness factors, the cardiovascular endurance and muscular strength were reported most important, and contribute to more efficient task execution in firefighting [14].

Kales et al. [15] reported that a minimum of 45 mL/kg/min of maximal oxygen uptake was required for firefighters to safely execute fire suppression task successfully, in the same manner, other studies addressed the need for at least 40 mL/kg/min of oxygen uptake capacity, and exercise capacity to perform at 84–100% of individual maximum heart rate [2,16]. The subjects who participated in our study showed a mean maximal heart rate of 186.3 bpm and VO_2_max of 54.7 mL/kg/min. In the correlation analysis between the circuit test time in the job-related performance test, the maximal heart rate (r = −0.735, *p* < 0.01) and VO_2_max (r = −0.739, *p* < 0.01) showed a high correlation. Circuit training represents an enduring and evolving training exercise format that was developed by R.E. Morgan and G.T. Anderson in 1953 at the University of Leeds in England [17].

The term circuit refers to a number of carefully selected exercises arranged consecutively. In the original format, 9–12 stations comprised the circuit; this number may vary according to the circuit’s design. Each participant moves from one station to the next with little (15–30 s) or no rest, performing a 15- to 45-s work bout of 8–20 repetitions at each station (using a resistance of about 40−60% of one-repetition maximum (1 RM)). The program may be performed with exercise machines, handheld weights, elastic resistance, calisthenics, or any combination of these. Circuit training is considered to be the main factor contributing to the high maximal heart rate and oxygen uptake as it is beneficial in cardiovascular endurance maintenance and development [18]. Furthermore, the study by Andrew et al. (2010) presented data at 95% of maximal heart rate of subjects on CPAT, in which our study developed the job-related performance test from, concurs with our reporting on the subjects’ heart rate.

On the other hand, our study conducted the Wingate test to examine the relationship of the anaerobic muscular power in firefighters. The Wingate test is employed universally to assess the anaerobic power by measuring the maximal pedaling speed onrmfl a cycle ergometer, where our study subjects exhibited a mean peak power of 7.62 W/kg, and mean average power of 5.67 W/kg. The correlation coefficient for each variable with the circuit test time for the job-related performance test was −0.484 (*p* < 0.05) for mean peak power, and −0.647 (*p* < 0.01) for mean average power.

Gledhill and Jamnik [7] and Von et al. [3] both reported the blood lactic acid concentration at about 6–13 mMol/L which is known to accumulate during intense physical activity with a high level of anaerobic metabolism. Although the study by Williams et al. [2] mentioned there was almost no known role played by anaerobic power in firefighting operations, our study along with Andrew et al. (2010) reported the circuit test time is closely related to anaerobic power. In addition to anaerobic power, our study also revealed a high correlation with blood lactate concentration, which on average marked at 15.7 mmol/L immediately after the test, 16.6 mmol/L, and 14.3 mmol/L at 5 min and 15 min into recovery. The correlation between the circuit test time was detected at each mark but was significant at 5 min (r = −0.450, *p* < 0.05) and 15 min (r = −0/460, *p* < 0.05).

Tomlin and Wenger [19] confirmed a high correlation between VO_2_max and blood lactate concentration recovery rate and identified that higher VO_2_max contributes to a faster recovery rate. The high correlation between the VO_2_max and recovery ratio identified in our study can be explained by those previous findings.

This study analyzed the relationship between VO_2_max, and anaerobic power, blood lactate concentration, and the current Korean firefighter physical fitness test. Analysis with the current test battery revealed high correlations with handgrip strength (r = −0.703, *p* < 0.01), low back strength (r = −0.681, *p* < 0.01), standing long jump (r = −0.618, *p* < 0.01), and shuttle run (r = −0.523, *p* < 0.01), while sit-ups showed a significant correlation (r = −0.397, *p* < 0.05). The sit and reach did not exhibit any correlation (*p* > 0.05). Michaelides et al. [13] stated abdominal strength, aerobic fitness, push-ups, and resting heart rate were significant predictors of firefighters’ task performance, and collectively with other previous studies, highlighted the importance of upper and lower body muscular strength owing to the tasks frequently performed at a fire scene such as carrying a victim, fire hose, or heavy object. Consequently, the job-related performance test suggested by this study showed a high correlation with muscular strength, agility, and cardiovascular endurance from the ongoing physical fitness test, and corresponds with main physical fitness factors mentioned in previous studies that looked at firefighters’ job-related performance [20,21]. These results suggest the good validity of the modified job-related physical fitness test as a measurement method for firefighters.

However, the subjects in our study exhibited top 15% physical fitness level out of 13,011 newly recruited male firefighters, which entails further research with a more diverse population in order to generalize our study’s modified job-related performance physical fitness test. Due to insufficient research on the physical fitness factors and the optimal firefighting task performance, we suggest the design of a test battery is required that can properly reflect the characteristics of a real-world fire scene.

## 5. Conclusions

This study modified the CPAT’s test battery to the Korean firefighter selection process and analyzed its validity to the firefighting task performance of 28 firefighter officer candidates. We conducted the modified job-related physical fitness test on subjects and recorded results of a portable blood gas analyzer, the Wingate test, and blood lactate test. Furthermore, we compared and analyzed the results from the ongoing physical fitness test as well. In consequence, the performance in modified job-related physical fitness tests revealed high correlations with VO_2_max, maximal heart rate, anaerobic peak power, blood lactate concentration recovery ratio, grip strength, low back strength, and shuttle run. The results are consistent with previous studies that analyzed fitness factors in firefighters and suggests a good level of validity of this modified test to assess firefighters’ job-related physical fitness factors.

## Figures and Tables

**Figure 1 ijerph-19-02587-f001:**
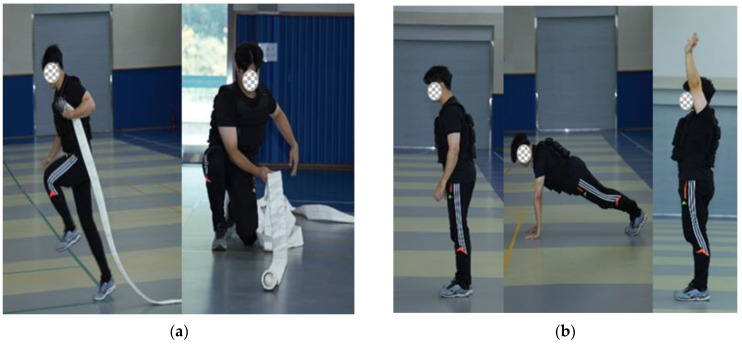
Changed physical fitness test sequence chart. (**a**) Burpee Test; (**b**) Hose drag & Pull; (**c**) Equipment Carry; (**d**) Medicine Ball Throw; (**e**) Inverted Row; (**f**) Sledge Push; (**g**) Tunnel get through; (**h**) Victim Drag; (**i**) 400 m Shuttle Run [10].

**Table 1 ijerph-19-02587-t001:** Characteristic subjects (*n* = 28).

Variables	Mean ± SD
Age (year)	29.1 ± 5.3
Height (cm)	173.4 ± 6.5
Weight (kg)	70.2 ± 8.9
Body Mass Index (kg/m^2^)	23.3 ± 2.4

**Table 3 ijerph-19-02587-t003:** List of measurement.

	Variables
Body composition	Height, Weight, Body mass index
Physical fitness	Muscle strength: Back and Handgrip strength Muscle endurance: Sit up Muscle power: Standing long jump Aerobic: Shuttle run, Gas respiration analysis Anaerobic power: Wingate test Flexibility: Sit and reach

**Table 4 ijerph-19-02587-t004:** Modified job related performance test measurement result.

Variables		Average Values	SD (*n* = 28)
Circuit Physical Test (s)		489.07	±78.97
VO_2_max	VO_2_max (mL/kg/min)	54.7	±6.5
	METS max	15.6	±1.9
	HR max (bpm)	186.3	±8.2
Lactate test	All out lactate value	15.7	±2.8
	Recovery (5 min) lactate value	16.6	±2.2
	Recovery (15 min) lactate value	14.3	±2.8
Anaerobic test	Peak power (W/Kg)	7.62	±1.16
	Avg power (W/Kg)	5.67	±0.80
	Power Drop (%)	57.4	±9.4
Korea the national firefighters physical fitness test	Grip strength (kg)	59.1	±7.5
	Low back strength (kg)	203.0	±29.7
	Sit and reach (cm)	25.3	±2.9
	Standing long jump (cm)	248.6	±18.8
	Sit up (rep/1 min)	49.3	±8.6
	Shuttle run (rep)	69.6	±12.3

**Table 5 ijerph-19-02587-t005:** Relationship of VO_2_max and Wingate test and lactate recovery ratio on correlation analysis.

	VM	ME	HM	AL	5RL	15RL	PP	AP	PD
CPT	−0.739 **	−0.735 **	−0.027	−0.109	−0.450 *	−0.460 *	−0.484 *	−0.647 **	0.360
ME			0.203	0.108	0.272	0.304	0.131	0.313	−0.309
HM				−0.095	0.076	0.069	−0.079	−0.040	−0.186
AL					0.711 **	0.686 **	0.220	0.221	0.018
5RL						0.928 **	0.487 *	0.530 **	−0.137
15RL							0.407 *	0.474 *	−0.279
PP								0.939 **	−0.027
AP									−0.199

** *p* < 0.01, * *p* < 0.05. Note: CPT = Circuit Physical Test Time, VM = VO_2_max, ME = METSmax, HM = Heart Rate Max, AL = All Out Lactate, 5RL= Recovery Lactate after 5 min, 15RL = Recovery Lactate after 15 min, PP = Peak Power, AP = Average Power, PD = Power Drop Ratio.

**Table 6 ijerph-19-02587-t006:** Relationship of improvement physical fitness test and Korea firefighter physical fitness test on correlation analysis.

	GS	LBS	SR	SLJ	SU	SR
CPT	−0.703 **	−0.681 **	0.184	−0.618 **	−0.397 *	−0.523 **
GS		0.847 **	−0.242	0.870 **	0.645 **	0.718 **
LBS			−0.392 *	0.759 **	0.521 **	0.586 **
SR				−0.076	−0.093	0.001
SLJ					0.559 **	0.830 **
SU						0.662 **

** *p* < 0.01, * *p* < 0.05. Note. CPT = Circuit Physical Test Time, GS = Grip Strength, LBS = Low Back Strength, SR = Sit and Reach, SLJ = Standing Long Jump, SU = Sit Up, SR = Shuttle Run.

## Data Availability

The data presented in this study are available on request from the corresponding author. The data are not publicly available due to privacy issues.

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
