# Peer review of "A Study on the Preliminary Validity Analysis of Korean Firefighter Job-Related Physical Fitness Test"

_ijerph, 2022, doi:10.3390/ijerph19052587_

Round 1

Reviewer 1 Report

Dear author,

I recommend you some minor changes before this manuscript can be published by the IJERPH.

Introduction:
In this section, I recommend you add a space, between “apparatus” and “(SCBA)” (line 39) and I suggest changing “health criterion” for “health criteria” (line 68).

Materials and methods section:
From my point of view, there is a lack of description in this section. Please, specify the materials and methods used for anthropometric measures (table 1) and for all the list of variables analyzed in your manuscript and described in tables, 2,3, and 4.
Besides, I suggest changing the “facts and fiqures” in table 2 for another most specific. And adding reference cites in each physical fitness test mentioned in Tables 2 and 3.

Results 
Please, I suggest changing “Average values” term for “Mean value” (table 5), in the same way, that it was used in table 1. Besides, specifying all acronyms used in the foot table note, according to the rule of the journal (It is applicable for all tables).

Discussion:
Please, introduce your discussion with the main outcome/s achieved of your study.
Furthermore, try to justify the statement mentioned in line 229 about the high level of cardiovascular endurance and muscular strength performances and decreasing of injury risk-related.

References: 
Please, delete the word “references” in reference number 1 (line 326).

Finally, I would like to suggest the use of horizontal separating lines in the same format in all tables of your manuscript according to the rules of the IJERPH, and please, you must cite properly, "et al.," in all cites of your manuscript.

Yours faithfully,

Author Response

attached file

Reviewer 2 Report

A Study on the Preliminary Validity Analysis of Korean Firefighter Job-related Physical Fitness Test were interesting. However, the introduction does not provide a sufficient background, it could be described in more detail. Especially the specificity of work for firefighters. The research design and methods are appropriate and have aroused the interest of the reader. The results could be made clearer. The conclusions are supported by the results. 

Author Response

attacted file

Reviewer 3 Report

Dear Authors

The article is important from a practical point of view, because the correct qualification of volunteers to work in the fire department and their ability to work during their professional activity is of key importance to ensure their safety and effectiveness. For this reason, the article is worth publishing. However, it is not a scientific paper and cannot be assessed in this respect. In the Introduction and Discussion the Authors did not take into account many new publications.

In the „Materials and Methods” lacks a detailed description of the study protocol. At what time interval were the volunteers tested taking into account the classic set of studies vs. modified set (how much time takes examinations, how long brake was between examinations according to standard protocol vs. modified one, if these examinations were performed in one day or  in different days etc.).

 In the „Conclusions”, the Authors should list the tests that they proposed in the modified version of the study protocol and indicate what benefits they see in the application of the modified protocol.

Detailed comments

Line 117  „This is referred to the weight of the equipment used in Korea based on ‘2020 Fire- fighting Equipment Standard Specification’ whereas the firefighters generally handle  gears such as a fire extinguisher, ladder, or rescue equipment”. I am not sure what  the Authors what the Authors wanted to say, that the weight of normally worn equipment is less / greater than that specified in this Specification? Please explain

Table 2 I am not sure what the term Victim returns means

Table 3 Inverted row – photo is unclear and it is difficult to understand what is the course of this  test, please explain

Line 257  „On the other hand, our study conducted the Wingate test to examine the relationship  of the anaerobic muscular power in firefighters” -  This sentence seems unfinished, because the relationship between which parameters should be indicated.

Author Response

attacted file

Reviewer 4 Report

I have had the opportunity to carefully read the article entitled: "A Study on the Preliminary Validity Analysis of Korean Firefighter Job-related Physical Fitness Test". The purpose of which was: "to review the job-related performance tests of overseas firefighters, evaluate their validity by comparing with the ongoing physical fitness test, and analyze the level of association with job performance in order to propose a Korean model firefighter physical fitness test".
The article has too many weaknesses to, in my opinion, be published in a prestigious journal. I point out the main points that should be improved in the future:

1.- The introduction should focus on the research question (objective), in a well-argued way with current literature, to establish the relevance of the question, and to understand the study design and methods used. What is the reason for modifying the already established firefighter selection tests? They argue about the time needed for selection of a large group of applicants, what is the advantage over the time taken for the current proposal, what is the specific difference between the work of an American or Canadian firefighter versus a Korean firefighter, what exactly do they intend to validate: the battery of tests versus specific physical fitness tests (in this case the association is the new one better than the old one?) or do they intend to validate the old battery versus the new battery? 

2.- The concept of validation of a test or a battery should be reviewed in more depth. Validating solely on the basis of the statistical significance of a correlation is a poor concept of validation. Studying the correlation between the tests of a battery only informs that they have a similar behavior. Moreover, they should analyze correlations not only on the basis of the p-value (which in general in their results is poor), analyze the coefficient of determination, do multivariate analysis,...  They should at least add adjectives to "validate".

3.- What permissions were obtained from the ethical point of view to carry out this research? It is not enough to sign a consent form.

4. How did you establish the sample size? What calculations did you make to establish the minimum sample size for validation?

5.- What was the selection process, what population does this sample represent?

6.- The statistical procedure is insufficiently explained, before performing the correlation, did you analyze the normality distribution?

7.- What is the physiological basis or the use or interpretations that can be obtained, (in the context of the "validation"), of quantifying blood lactate (they should indicate in the results the units of lactatemia), before, 5 and 15 minutes after the proposed tests? It is assumed but not clearly expressed that they determined the VO2max not in the laboratory but by monitoring while doing the Shuttle Run test, which is already sufficiently contrasted to estimate the Vo2max, so the direct determination by this test does not contribute much to the work, (what difference did they find between the measured VO2max and the estimated by the test?)

8.- In the presentation of the results express the maximum peak oxygen in ml/kg/min or in METs, but not in both, it is an obvious repetition. why do they indicate the maximum heart rate?  In the results they only show weight and height, why do they indicate that they used the InBody 720 and did not use and analyze the different variables that the exploration with bioimpedance yields? 

9.- The methodology used should not be repeated in the conclusions. The conclusions formulated do not really respond to the objectives expressed in the introduction of the study. On the other hand, it is obvious that all physical condition tests correlate with each other: in a generic population there is always a statistically significant correlation between physical tests, or between laboratory tests of physical qualities and physical tests. 

10.- The bibliography used is a little limited, not very updated, and presents many errors (always point out the volume, volume, start and end page... and in the electronic ones the publication identifier if it does not have volume and pages, which in your case it does have and you have not put.

I recognize their effort, but I observe the methodological limitations of the work that should be improved in the future.

Author Response

attacted file

Round 2

Reviewer 4 Report

Dear authors: the usual practice is to respond in a reasoned manner to EACH of the reviewers' comments, indicating what has changed and how the change has been made.

Neither have they responded to my observations (in one sense or another), nor do I see substantial changes that would induce me to change my opinion on the article submitted.

Author Response

Dear reviewer!

First of all, in your busy schedule, you reviewed the artcle, and you gave your opinion on the parts to be corrected. I'm really sorry.

The contents that need to be supplemented among the contents of the review opinion have been supplemented as much as possible. Thank you for your understanding.

While humbly accepting the reviewers' opinions on whether the necessity of the research, the adequacy of the research design, and the conclusion of the research are adequately described, nonetheless, prior research on the physical fitness test and qualification of fire-fighters in Korea Please understand that it may be somewhat insufficient as there are not many of them to accommodate.

While humbly accepting the reviewers' opinions on whether the necessity of the study, the adequacy of the study design, and the conclusion of the study are adequately described, nonetheless, prior research on the physical fitness test and qualification of firefighters in Korea Please understand that it may be somewhat insufficient as there are not many of them to accommodate.
